# Study on Cavitation, Warpage Deformation, and Moisture Diffusion of Sop-8 Devices during Molding Process

**DOI:** 10.3390/mi14122175

**Published:** 2023-11-29

**Authors:** Wenchao Tian, Shuaiqi Zhang, Wenbin Li, Yuanming Chen, Jingrong Zhao, Fei Xin, Yingying Qian, Wenhua Li

**Affiliations:** 1Key Laboratory of Electronic Equipment Structure Design (MOE), School of Mechano-Electronic Engineering, Xidian University, Xi’an 710071, China; 22041212704@stu.xidian.edu.cn (S.Z.); wenbinli99@stu.xidian.edu.cn (W.L.); fxin@xidian.edu.cn (F.X.); 21041211881@stu.xidian.edu.cn (Y.Q.); wenhuali@stu.xidian.edu.cn (W.L.); 2Sharetek Industrial Equipment Co., Ltd., Shanghai 201109, China; frankchen@sharetek.com.cn (Y.C.); jr.zhao@sharetek.com.cn (J.Z.)

**Keywords:** plastic packaging, injection molding, cavitation, warpage deformation, moisture diffusion

## Abstract

Plastic packaging has shown its advantages over ceramic packaging and metal packaging in lightweight, thin, and high-density electronic devices. In this paper, the reliability and moisture diffusion of Sop-8 (Small Out-Line Package-8) plastic packaging devices are studied, and we put forward a set of complete optimization methods. Firstly, we propose to improve the reliability of plastic packaging devices by reducing the amount of cavitation and warpage deformation. Structural and process factors were investigated in the injection molding process. An orthogonal experiment design was used to create 25 groups of simulation experiments, and Moldflow software was used to simulate the flow mode analysis. Then, the simulation results are subjected to range analysis and comprehensive weighted score analysis. Finally, different optimization methods are proposed according to different production conditions, and each optimization method can reduce cavitation or warpage by more than 9%. The moisture diffusion of the Sop-8 plastic packing devices was also investigated at the same time. It was determined that the contact surface between the lead frame and the plastic packaging material was more likely to exhibit delamination under the condition of MSL2 moisture diffusion because the humidity gradient was easily produced at the crucial points of different materials. The diffusion of moisture is related to the type of plastic packaging material and the diffusion path.

## 1. Introduction

With the rapid development of integrated circuits, semiconductor chips are ubiquitous. Therefore, the reliability of devices, especially in high-precision and military fields, has put forward higher requirements [1,2]. Packaging forms of semiconductor chips can be divided into plastic packaging, ceramic packaging, metal packaging, etc. [3]. Compared with plastic packaging, ceramic packaging has a higher process temperature, cost, and brittleness and is prone to stress damage. Additionally, the ceramic packaging industry has high technological barriers in terms of research development and production, so the price of ceramic packaging devices is relatively high [4]. Metal packaging has the characteristics of high mechanical strength and excellent heat dissipation performance, but the disadvantages of metal and chip thermal expansion coefficient differences, large quality, high density, high cost, less flexibility, and so on, limit the application of metal packaging in the field of high-end products. Although plastic packaging is inferior to ceramic packaging in terms of heat dissipation, heat resistance, and tightness, plastic packaging has the advantages of low cost, a thin shape, a simple process, and being suitable for automated production [5,6]. Plastic packaging is the most popular method in the microelectronics industry. It is widely employed in various equipment, from standard consumer electronics items to precise ultra-high-speed computers [7].

Three types of failure frequently occur in plastic packaging devices during actual production: warpage deformation, cavitation, and delamination. Warpage deformation refers to the deformation of the plastic packaging devices after the demolding treatment, mainly because of the unreasonable design of the injection mold or the mechanical properties of the material, which will reduce the reliability of the devices [8,9]. Cavitation refers to the phenomenon that there will be holes on the surface of the plastic packaging devices after the demolding treatment, which is because the air or moisture cannot be eliminated during the injection molding process of the equipment and remains in the cavity to become bubbles. Sink marks are the depressions and unevenness of the device surface. Sink marks are the heat-induced shrinkage of the plastic surface due to loss of pressure during the injection molding process. Voids are defects in the device that resemble vacuum bubbles. Voids are present because the bending strength of the device surface after cooling is greater than the stress strength of the internal contraction, and the device forms voids inside the shrinkage process. Sink marks and voids are the same, and they are collectively referred to as cavitation in this paper. The Venturi effect describes how the velocity of a fluid increases as the cross-section of the container it flows in decreases. As the fluid with the higher speed enters the smaller cross-section, it creates a vacuum and gives momentum to the other fluid [10,11,12]. The cavitation failure diagram is shown in Figure 1. Too many large cavitations will reduce the reliability of the devices. Delamination refers to the delamination phenomenon at the interface between the plastic layer and other materials of the plastic packaging devices, which can cause performance degradation or even failure of the devices [13].

In 2019, Rovitto et al. used Moldflow (2018) to simulate the process of epoxy resin injection and encapsulation of microchips using the mold flow analysis method. The study found that the unbalanced flow behavior of resin in the mold cavity led to the formation of cavitation in the mold packaging [14]. In 2020, Grandi et al., through process modeling and the finite element analysis method, studied the influence of different plungers on defects such as cavitation, weld marks, and gold wire offset caused by resin flow in the TQFP (Thin Quad Flat Package) chip packaging process [15]. Shi et al. published a study in 2021 that looked at the variation of moist thermal stress caused by moisture diffusion in QFN (Quad Flat No-leads Package) plastic packaging devices, proposed an analysis method for continuous moisture diffusion under variable temperature conditions, and established the corresponding relationship between wet-mechanical stress parameters and steam stress. The formula for total wet and heat stress is then derived using the superposition principle. The results demonstrate that this method can be used to analyze water diffusion and stress under dynamic temperature and non-uniform humidity distributions [16]. Uschitsky M. et al. investigated the effect of water diffusion on the mechanical properties of various epoxy sealant compounds after moisture absorption in 2001. The results show that the degree of water diffusion is primarily determined through the sample’s relative humidity and the concentration of the filler. The wet thermal stress caused by the filler is relatively low, but the compound’s strength is significantly reduced, and its plasticity is significantly increased [17]. The reliability analysis of plastic packaging devices at home and abroad still mainly depends on simulation analysis, and the validity of simulation analysis has been confirmed. However, the present research mainly studies a single factor or combination of a few factors on the reliability of plastic packaging devices, and there is still a lack of a research method that can systematically consider the influence of structure and process on plastic packaging devices in the production process.

To sum up, plastic packaging is the most popular form of packaging in the microelectronics industry, and the mold structure and plastic packaging process are the main causes of failures in plastic packaging devices. Therefore, this paper takes the Sop-8 (Small Out-Line Package-8) devices provided by a cooperative enterprise as the research object. An optimization method is proposed to improve the reliability of plastic packaging devices, which can comprehensively consider various influencing factors. This method uses Moldflow and ANSYS (ANSYS_2020R2) and other software; adopts an orthogonal experimental design method, range analysis method, comprehensive weighted analysis method, and other methods; puts forward better technological parameters and structural parameters in the plastic packaging process; and analyzes the influence degree of various factors [18,19,20,21,22]. This method is a universal method that can effectively select the optimal scheme and solve the same kind of problems. Moisture diffusion of plastic packaging devices was also investigated, and moisture diffusion simulation tests were performed on plastic packaging devices [23]. The main reason for the delamination of devices is clarified, and the influence of humidity gradient, chip size, plastic packaging material, and other factors on moisture diffusion is studied. Finally, an optimization method is proposed for device delamination.

## 2. Viscosity Model and Moisture Diffusion Theory

### 2.1. The Herschel–Bulkley–WLF Viscosity Model

The reactive viscosity model is used to represent the correlation of temperature, shear rate, and curing behavior during the curing of thermosetting materials. In Moldflow, this model can be used for reaction molding analysis, microchip package analysis, and underlying overcoat package analysis. The viscosity of a polymer can be expressed using the following equation.
(1)$$\eta =\frac{\tau_{y}}{\gamma }+K\gamma^{\left(n-1\right)}={\left(\frac{K\tau^{\left(n-1\right)}}{1-\frac{\tau_{y}}{\tau }}\right)}^{1/n}$$
(2)$$\tau_{y}=\tau_{y}{}_{0}\exp\left(\frac{T_{y}}{T}\right)$$

In Equations (1) and (2), η is the viscosity of the polymer, the unit is Pa·s; γ is the shear rate of the polymer, the unit is 1/s; K denotes the WLF viscosity, τ denotes the shear stress, and T is the absolute temperature in *K*. Moreover, n, $\tau_{yo}$ and $T_{y}$ are associated constants.

The Williams–Landel–Ferry model (WLF) is commonly used to melt polymers or other liquids with glass transition temperatures and is usually expressed as follows:(3)$$K=K_{0}\exp\left[-\frac{C_{a}(T-T_{g})}{C_{b}+(T-T_{g})}\right]$$
(4)$$K_{0}=K_{00}{\left(\frac{\alpha_{g}}{\alpha_{g}-\alpha }\right)}^{\left(C_{1}+C_{2}\alpha \right)}$$

In Equations (3) and (4), α denotes the curing degree of the polymer, also known as the conversion rate, and its value is between 0 and 1; $C_{a}$, $C_{b}$, $C_{1}$, $C_{2}$, $K_{00}$, and $T_{g}$ are the data fitting coefficients.

### 2.2. Theory of Moisture Diffusion

The diffusion of moisture into the epoxy plastic sealing material is a slow process, which is the result of the diffusion and movement of water molecules from the high to low moisture region. The diffusion of moisture can be expressed using Fick’s second law, which describes the change of concentration at each point in the medium due to diffusion under unstable diffusion conditions; that is, the diffusion rate of water molecules is proportional to the second derivative of its concentration, as follows:(5)$$\frac{\partial C}{\partial t}=D\frac{\partial^{2}C}{\partial x^{2}}$$

In the above equation, C is the volume concentration of the diffusing substance, the unit is (kg/m^3^); t is the diffusion time, the unit is s; and x is the distance, the unit is m. D is the diffusion coefficient, the unit is mm^2^/s. The wet conductivity of a material is defined as the product of the moisture diffusion coefficient $\left(D\right)$ and the saturation moisture $\left(C_{sat}\right)$. In the actual situation, D varies with the change of concentration, and it is usually treated approximately as a constant to facilitate the solution of the diffusion equation.

For an isotropic three-dimensional diffusion system, that is, $D_{xx}=D_{yy}=D_{zz}$, the formula for moisture diffusion is as follows:(6)$$\frac{\partial C}{\partial t}=D\left(\frac{\partial^{2}C}{\partial x^{2}}+\frac{\partial^{2}C}{\partial y^{2}}+\frac{\partial^{2}C}{\partial z^{2}}\right)$$

## 3. Cavitation and Warpage Deformation

In the plastic packaging process, the injection molding process is the most important. The injection molding is to place the chip and lead frame bonded by the lead wire in the mold cavity, and then inject the molten plastic compound to wrap the wafer and the gold wire on the lead frame. This can protect the element from damage, prevent the gas from oxidizing the internal chip, and ensure the safety and stability of the product. Therefore, improving the yield and reliability of plastic packaging devices requires optimizing the injection molding process. Process parameters and structural parameters are significant factors that influence the injection molding process. Therefore, this paper takes the following part of simulation optimization as an example and puts forward an optimization process suitable for the reliability optimization of similar plastic packaging devices. The optimal injection molding process parameters and structural parameters were proposed to optimize the injection molding process [24,25,26]. The relationship between cavitation number and warping deformation is discussed to guide production practice.

### 3.1. Initial Model Simulation

The 3D models of the Sop-8 lead frame, chip, die, and cavity were created with Solid Works (SW2019) using the engineering drawings that a company provided as a source of reference. Moldflow then meshes the model, and the material parameters are provided. Finally, the boundary conditions such as constraints, exhaust ports, and injection ports are added, and the simulation analysis is performed [27]. In the following, a single base island frame simulation is taken as an example.

Simulation modeling is carried out as shown in Figure 2, where Figure 2a is modeling the 3D model of the lead frame and Figure 2b is modeling the 3D explosion diagram of the model.

The model is meshed using Moldflow. Then, the boundary conditions are set, and six degrees of freedom are fixed on the nodes of all the surfaces of the upper and lower molds. In addition, 40 exhaust ports in total are placed in the cavity. Thermosetting plastic packaging material was selected as the plastic packaging material. Table 1 shows the coefficient of the reactive viscosity model obtained using Origin curve fitting after testing the correlation between temperature, shear rate, and curing behavior in the curing process of materials [28,29,30]. Table 2 and Table 3 show the initial process parameters and the initial structural parameters, respectively [31].

The simulation results show that under the conditions of initial process combination parameters, the maximum warpage deformation of a single base island frame device is 0.0203 mm, and the amount of cavitation is 26. 

### 3.2. Orthogonal Experiment Design

The process parameters were divided into 25 groups according to the orthogonal experimental design, as shown in Table 4. Simulation analysis was carried out on the initial structure parameter combination, and the warpage deformation and cavitation amount of each group of experimental devices were counted. Because the reliability of the devices is closely related to the amount of cavitation in the plastic packaging body in actual production, the optimal process combination of cavitation is selected using the range analysis, and then the structural parameter optimization simulation is carried out. Based on the optimal cavitation process combination, the structural parameters were divided into 25 groups according to the orthogonal design experiment method and simulated using Moldflow, and the warpage deformation and the amount of cavitation were calculated, as shown in Table 5.

### 3.3. Results and Discussion

According to the two parts of simulation data obtained and the experience in the actual production process, three different optimization schemes are proposed for the process parameters and structural parameters of the Sop-8 plastic packaging devices under the framework of a single base island. They are Optimal Warpage (Optimal parameter combinations with minimal warpage), Optimal Cavitation (Optimal parameter combinations with minimal cavitation), and Optimal Comprehensive (Optimal parameter combinations under comprehensive weighted score analysis).

When only the influence of the warpage deformation or the amount of cavitation on the devices is considered, the Optimal Warpage and the Optimal Cavitation are obtained using the range analysis. When the influence of warpage deformation and the amount of cavitation on the devices need to be considered comprehensively, the Optimal Comprehensive is obtained via scoring according to the weight of 3:7.

#### 3.3.1. Process Parameter Optimization

After analyzing the obtained data, the optimization table of process parameters is shown in Table 6.

Through the range analysis, when only the influence of process parameters on the warpage deformation is considered, the ranking of factors that have the greatest influence on the warping deformation of the devices can be obtained: Mold temperature > Melt temperature > Injection time > Curing time > Injection pressure. When only the influence of process parameters on the amount of device cavitation is considered, the ranking of factors that have the greatest influence on the amount of device cavitation can be obtained: Melt temperature > Injection pressure > Injection time > Mold temperature > Curing time. All three schemes can significantly reduce the amount of cavitation, and only the Optimal Warpage can reduce the warpage deformation by about 16.3%.

#### 3.3.2. Structural Parameters Optimization

After analyzing the obtained data, the optimization table of structural parameters is shown in Table 7.

When only the influence of structural parameters on warpage deformation is considered, the ranking of factors that have the greatest influence on the warpage deformation of the devices can be obtained through the range analysis: Width of Flow channels = Chamfer > Height of Gates > Height of Flow channels > Width of Gates > Width of Exhaust ports > Height of Exhaust ports. When only the influence of structural parameters on the amount of device cavitation is considered, the ranking of factors with the greatest influence on the amount of device cavitation can be obtained: Height of Gates = Width of Gates > Width of Exhaust ports > Height of Exhaust ports > Height of Flow channels > Width of Flow channels = Chamfer. All three schemes can significantly reduce the amount of cavitation in the devices. The Optimal Warpage and the Optimal Comprehensive can reduce the warpage deformation of the devices by about 9.85% and 5.42%, respectively, but the Optimal Cavitation will increase the warpage deformation of the devices by about 108.3%. The above conclusion is only for the scoring index, but in the actual process of structure production and manufacturing, it is necessary to consider the actual situation.

From the simulation results, it can be seen that the Melt temperature and Injection pressure have a large impact on the number of cavitations in the device, while the Melt temperature is selected as a lower value and the Injection pressure is selected as a larger value. This is related to the principle of the formation of cavitation. Too high a temperature will make the temperature difference between the inside and outside of the device too large, and it is easier to generate stress, thus causing the surface to shrink and collapse. However, the Injection pressure is related to the Holding pressure. The increase in the Holding pressure will reduce the number of cavitations.

According to the data feedback from the cooperative enterprise and the third-party test report, the simulation scheme can improve the yield and reliability of plastic sealing devices. Figure 3a shows the acoustic sweep after the high-temperature steam test of the initial group, and Figure 3b shows the acoustic sweep after the high-temperature steam test of the optimization group.

The actual production feedback of the cooperative enterprise according to the optimized scheme shows that the optimized simulation scheme can improve the yield and reliability of plastic packaging devices. The optimization process has high optimization efficiency and reliability. The process is as follows: Decompose Impact Factor–Orthogonal Experiment Design–Range Analysis–Comprehensive Weighted Analysis. The most representative test combination is selected from many test schemes using the planned orthogonal table, the test scheme is reasonably arranged, and the minimum number of tests is designed to understand the influence of various factors on the test index. Due to the orthogonality and comprehensive comparability of the Orthogonal Experiment Design, we can compare this test’s primary and secondary factors more intuitively using the Range Analysis and find out the optimal level of collocation through simple comparison. In each orthogonal test, it is found that in multiple index evaluation tests, each factor and its corresponding level have different effects on the two indexes of device warping deformation and device cavitation number. When one index improves, another index may decrease. The Comprehensive Weighted Analysis method is used to determine the index’s corresponding weight or coefficient according to the importance of each index and to comprehensively score the index to obtain the parameter combination considering multiple test indexes.

## 4. Moisture Diffusion

Plastic packaging has the advantages of low cost, a thin shape, a simple process, and being suitable for automatic production. However, plastic packaging is sensitive to humidity, and the dielectric constant of the material can be changed due to the external water vapor. So, there will often be delamination failure in different positions [32,33]. In this paper, the Sop-8 devices are selected as the research object, and ANSYS is used to study the distribution of moisture at different locations and the diffusion law of moisture inside the devices under a hygrothermal environment. This paper mainly studies the deformation and stress of SOP-8 plastic packaging devices under the moisture absorption condition of moisture sensitivity class 2 under the IPC/JEDEC J-STD-020D.1 standard [34]. In the following analysis, the moisture sensitivity class 2 test is referred to as MSL2 [35,36].

### 4.1. Simulation Model

Solid Works (2019) was used to model the devices in 3D. The modeling part mainly includes a plastic shell, Cu lead frame, Si chip, and solder, which adopts the eutectic welding method to fix the chip on the lead frame [37]. Figure 4 shows the device’s structure diagram. Figure 5 shows the device dimensions.

Then, the material parameters are given, and the mesh is divided. The materials in the simulation of wet gas diffusion are isotropic materials, which require thermodynamic parameters and related wet parameters [38,39,40]. The wet parameters include saturated moisture and wet conductivity. The dimensions of plastic packaging material and lead frame elements are set to 0.1 mm, and the dimensions of Si chip and solder elements are set to 0.75 mm. The lead frame part uses Multizone division, and the soldered part is divided into two layers using Edge Sizing. The material thermodynamic parameters are shown in Table 8, and the material wet parameters are shown in Table 9.

Finally, load and boundary conditions are added, in which the boundary load of the MSL2 thermal simulation is 85 °C, the relative humidity load is 60%, and the temperature and humidity load last for 168 h. A fixed constraint is applied to the bottom of the eight pins of the SOP-8 plastic packaging devices. Since the lead frame, solder, and Si chip almost do not absorb moisture, and ANSYS cannot set zero parameters, the outer surface of the lead frame, solder, and Si chip are all set with absolute moisture boundary conditions.

### 4.2. Results and Discussion

Through the simulation experiment, it can be seen that after the end of the moisture diffusion test, the moisture absorption of the devices reaches the saturation state, but there is a large humidity gradient at the critical position of different materials [41]. The existence of the humidity gradient seriously affects the quality of packaged devices. On the one hand, it affects the elastic modulus and strength of the material at high temperatures; on the other hand, it corrodes the internal metal layer of the devices.

Five points A–E at the critical positions of the plastic packaging material, lead frame, and Si chip are selected to analyze the moisture diffusion at different interface positions in the package body, as shown in Figure 6.

The plastic packaging devices continuously absorb the external moisture and gradually diffuse from the surface of the devices to the interior. As the time passes, the diffusion rate gradually decreases and finally reaches the moisture absorption saturation state smoothly, which is consistent with the moisture diffusion behavior predicted using the Fick diffusion law [42,43]. The ranking of the speed of change is D > E > A > B>C.

In the process of moisture diffusion, the wet expansion coefficient between various materials is mismatched due to temperature differences, and the wet stress similar to thermal stress will be generated inside the devices due to the moisture expansion of the humidity gradient. The overall stress of each structure gradually decreases from the periphery to the center, as shown in Figure 7a. In the process of continuous moisture absorption and expansion, the internal part of the devices will be continuously compressed, and the delamination between the plastic packaging material and the lead frame, solder, and the contact surface of the Si chip will happen under the action of wet stress. The overall deformation of each structure will gradually decrease from around to the center, as shown in Figure 7b.

Finite element analysis software (Ansys 19.2) was used to calculate the total deformation of the five dangerous points on the contact surface of different materials [44]. The total deformation point is D > E > A > B > C. Therefore, compared with the contact surface between the chip and the plastic packaging material, the contact surface between the pin of the lead frame and the plastic packaging material is more likely to appear as delamination in the MSL2 simulation [45].

### 4.3. Si Chip Size

The delamination of the devices is due to the delamination between the plastic packaging material and the frame, solder, and chip. From the simulation results of the five danger points, it can be conjectured that the moisture gradient, diffusion velocity, wet stress, and wet strain are related to the length of the path from outside the devices to the chip. Therefore, the influence of path length on moisture diffusion will be studied in the following. Moreover, due to the hydrophilicity of plastic packaging material, the too-large size of the plastic body will make the overall deformation of the device larger, and the extrusion of external pins will cause contact failure, so we only increase the size of the chip, and the size of the external plastic body will remain unchanged. The chip size table is shown in Table 10.

The total deformation of moisture and heat superposition of the above five danger points are obtained, and the influence of different sizes on the total deformation of moisture and heat superposition is analyzed. The total deformation of moisture and heat superposition table is shown in Table 11.

In the case of other moisture diffusion simulation settings being unchanged, only changing the chip size, the maximum deformation of five points from large to small is D > E > A > B > C, the maximum deformation is point D, and the minimum deformation is point C, which is consistent with the above analysis, so point D is regarded as a dangerous point. The order of deformation from large to small is 4 > 2 > 3 > 1> initial size. The deformation of point D from large to small is 6.2559 μm, 6.1524 μm, 5.3179 μm, 4.5177 μm and 4.4902 μm. The relationship between the chip size and the maximum deformation is not linear, which first increases, then decreases, and then increases. In general, the maximum total deformation increases gradually with the increase in chip size. Among the five chip sizes, the minimum deformation is the initial size, and the maximum is size 4. Because point D, among the five points A–E, has the shortest path from the external environment. As the chip size increases, the path between points A and B inside the devices and the external environment gradually decreases, and the total deformation of moisture and heat superposition gradually increases due to the influence of moisture diffusion. Point C is located in the center of the devices and is less affected by the chip size transformation. Therefore, the small size of the chip can reduce the delamination effect caused by moisture diffusion.

Increasing the elastic modulus of plastic packaging materials and reducing the saturated moisture degree of materials can reduce the superposition deformation of moisture and heat in plastic packaging devices, thereby reducing the extrusion effect of moisture diffusion on device pins and the influence of delamination on device reliability.

Therefore, we should choose a plastic packaging material with a large elastic modulus and small material saturation moisture. At the same time, the size of the plastic body should be expanded, and the size of the chip and solder should be reduced without increasing the influence of the expansion of the plastic body on the extrusion of the pin.

## 5. Conclusions

In this paper, based on finite element analysis theory, Moldflow (2018) software was used to analyze the flow mold in the injection molding process, and the orthogonal experimental design method, range analysis, and comprehensive weighting score method were used to find out the parameter combinations of process parameters and structural parameters in three different cases. The moisture diffusion of plastic packaging devices is studied, which provides a theoretical basis for the device’s delamination failure. And the following conclusions are drawn: For the process parameters, all three schemes can significantly reduce the amount of cavitation, and only the Optimal Warpage can reduce the warpage deformation by about 16.3%; for the structure parameters, all three schemes can significantly reduce the amount of cavitation of the devices. The Optimal Warpage and the Optimal Comprehensive can reduce the warpage deformation of the devices by about 9.85% and 5.42%, respectively, but the Optimal Cavitation will increase the warpage deformation of the devices by about 108.3%. In the actual production process, it is necessary to consider not only the influence of warping deformation and the amount of cavitation but also the economic cost and manufacturing cycle, and finally, choose the scheme that can improve the production yield and the reliability of the devices.This paper proposes a universal method that can effectively select the optimal scheme. The optimal scheme can be obtained by decomposing the impact factors to design Orthogonal experiments and then analyzing the data using range analysis and comprehensive weighted analysis. Adopting more efficient materials or frame structures will affect the experimental results, but the optimal process parameters and structural parameters under the new model conditions can still be obtained according to the above test method.Under the condition of MSL2 moisture diffusion, the critical position of different materials easily produces the humidity gradient, and the contact surface between the lead frame and the plastic packaging material is more likely to appear delaminated. It is also verified that the moisture gradient is related to the length of the diffusion path and the types of plastic packaging material.This paper verifies the feasibility of using finite element analysis for process and structural simulation and moisture diffusion.Increasing Young’s modulus of the plastic packaging materials and reducing the saturated humidity of the materials can reduce the hygrothermal superposition deformation of the plastic packaging devices, which can reduce the impact of moisture diffusion on the extrusion of the device pins and the impact of delamination on the reliability of the devices. Therefore, the plastic packaging material with high Young’s modulus and low material saturation humidity should be selected, and at the same time, the size of the plastic packaging body should be enlarged, and the size of the chip and solder should be reduced without increasing the effect of the expansion of the plastic packaging body on the extrusion of the pins.The advantages of plastic packaging are irreplaceable for ceramic packaging and metal packaging. The optimization of the plastic packaging process has a certain limit on the reliability improvement of the device. Therefore, we must make breakthroughs in plastic packaging materials to contribute to the whole plastic packaging industry.

## Figures and Tables

**Figure 1 micromachines-14-02175-f001:**
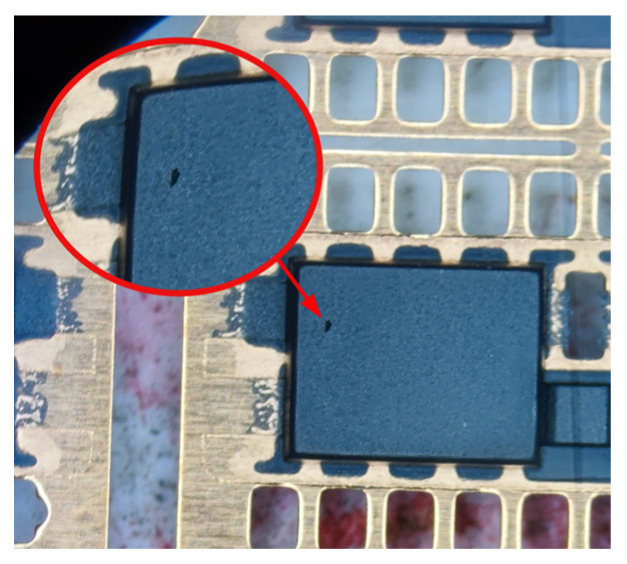
Cavitation failure diagram.

**Figure 2 micromachines-14-02175-f002:**
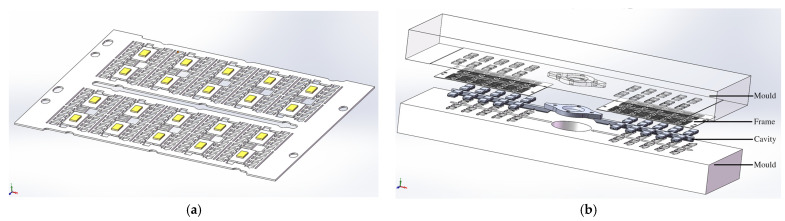
Three-dimensional simulation model. (**a**) Three-dimensional model of the lead frame; (**b**) Three-dimensional explosion diagram of the model.

**Figure 3 micromachines-14-02175-f003:**
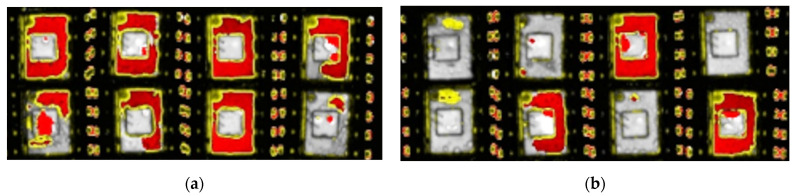
Acoustic sweep after high-temperature steam test. (**a**) Initial scheme experiment; (**b**) Optimized scheme experiment.

**Figure 4 micromachines-14-02175-f004:**
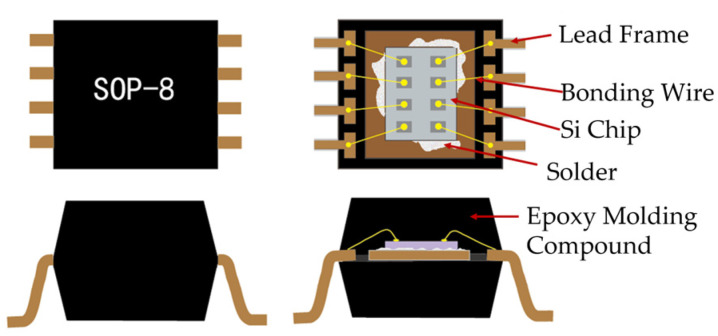
SOP-8 structure diagram.

**Figure 5 micromachines-14-02175-f005:**
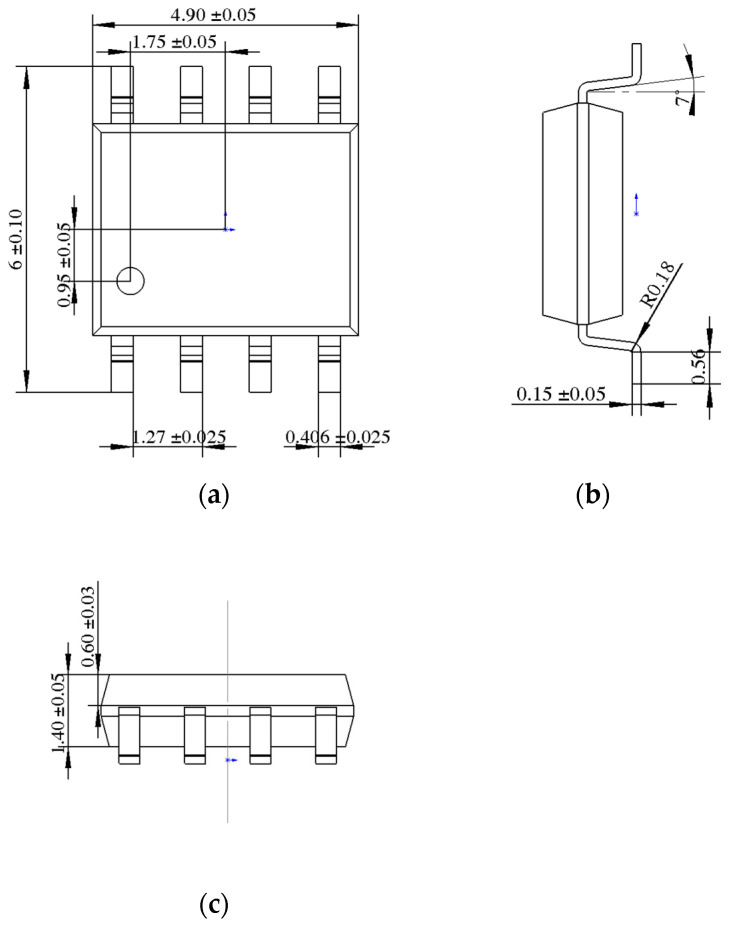
SOP-8 devices dimensions. (**a**) Front view; (**b**) Right view. (**c**) Top view.

**Figure 6 micromachines-14-02175-f006:**
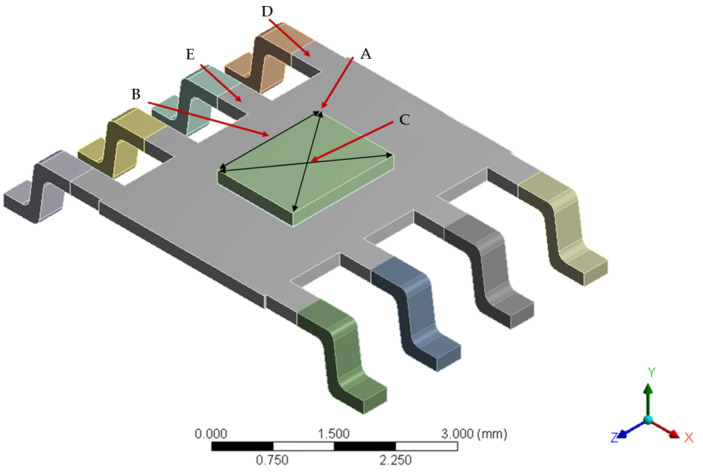
Node location of SOP-8 plastic packaging devices.

**Figure 7 micromachines-14-02175-f007:**
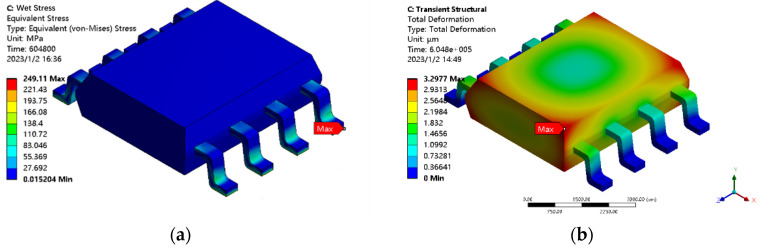
Wet stress and wet deformation diagrams of SOP-8 plastic packaging devices. (**a**) Wet stress diagram of SOP-8 plastic packaging devices; (**b**) Wet deformation diagram of SOP-8 plastic packaging devices.

**Table 1 micromachines-14-02175-t001:** Coefficients of the reactive viscosity model.

Parameters	Data	Units
n	0.5838	—
Tau	0.0001	Pa
B	302.822	Pa·s
Tb	2300.13	K
c1	3.762	—
c2	10.22	—

**Table 2 micromachines-14-02175-t002:** Initial process parameters.

Preheating Time (s)	Curing Time (s)	Injection Pressure (kg/cm²)	Injection Molding Time (s)	Mold Clamping Pressure (kg/cm²)
5	150	36	12	31

**Table 3 micromachines-14-02175-t003:** Initial structural parameters (Units: mm).

Height of Gate	Height of Flow Channel	Height of Exhaust Ports	Width of Gate	Width of the Flow Channel
0.04	0.04	0.04	1.17	1.1

**Table 4 micromachines-14-02175-t004:** Process parameters orthogonal experiment design and results.

Factor Classify	Factor A	Factor B	Factor C	Factor D	Factor E	Warpage Deformation (mm)	Cavitation
	Melt Temperature (Units: °C)	Mold Temperature (Units: °C)	Inject Time (Units: s)	Curing Time (Units: s)	Injection Pressure (Units: kg/cm^2^)		
1	155	155	8	130	24	0.0151	25
2	155	165	12	160	48	0.0177	23
3	155	175	16	140	42	0.0202	22
4	155	185	10	170	36	0.0223	21
5	155	195	14	150	30	0.0275	22
6	165	155	16	160	36	0.0161	27
7	165	165	10	140	30	0.0183	26
8	165	175	14	170	24	0.0203	23
9	165	185	8	150	48	0.0228	32
10	165	195	12	130	42	0.0249	22
11	175	155	14	140	48	0.0164	27
12	175	165	8	170	42	0.0214	27
13	175	175	12	150	36	0.0203	26
14	175	185	16	130	30	0.0227	27
15	175	195	10	160	24	0.0256	28
16	185	155	12	170	30	0.0168	26
17	185	165	16	150	24	0.0186	24
18	185	175	10	130	48	0.0207	27
19	185	185	14	160	42	0.0232	25
20	185	195	8	140	36	0.0249	25
21	195	155	10	150	42	0.0153	23
22	195	165	14	130	36	0.0278	28
23	195	175	8	160	30	0.0254	25
24	195	185	12	140	24	0.0346	27
25	195	195	16	170	48	0.0268	26

**Table 5 micromachines-14-02175-t005:** Structure parameters orthogonal experiment design and results.

Factor Classify	Factor A	Factor B	Factor C	Factor D	Factor E	Factor F	Factor G	Warpage Deformation	Cavitation
	Height of Gates	Height of Flow Channels	Height of Exhaust Ports	Width of Gates	Width of Flow Channels	Width of Exhaust Ports	Chamfer		
1	0.025	0.025	0.025	0.9	0.9	0.9	0.01	0.0445	22
2	0.025	0.035	0.035	1	1	1	0.015	0.0344	29
3	0.025	0.05	0.05	1.1	1.35	1.1	0.02	0.0297	31
4	0.025	0.1	0.1	1.2	1.6	1.2	0.025	0.0187	34
5	0.025	0.15	0.15	1.3	1.85	1.3	0.03	0.0209	22
6	0.035	0.025	0.035	1.1	1.6	1.3	0.025	0.0203	25
7	0.035	0.035	0.05	1.2	1.85	0.9	0.03	0.0195	32
8	0.035	0.05	0.1	1.3	0.9	1	0.01	0.0368	25
9	0.035	0.1	0.15	0.9	1	1.1	0.015	0.0273	25
10	0.035	0.15	0.025	1	1.35	1.2	0.02	0.0212	32
11	0.05	0.025	0.05	1.3	1	1.2	0.015	0.0429	26
12	0.05	0.035	0.1	0.9	1.35	1.3	0.02	0.0341	19
13	0.05	0.05	0.15	1	1.6	0.9	0.025	0.02	28
14	0.05	0.1	0.025	1.1	1.85	1	0.03	0.0211	28
15	0.05	0.15	0.035	1.2	0.9	1.1	0.01	0.0238	24
16	0.1	0.025	0.1	1	1.85	1.1	0.03	0.0199	21
17	0.1	0.035	0.15	1.1	0.9	1.2	0.01	0.04	25
18	0.1	0.05	0.025	1.2	1	1.3	0.015	0.0248	25
19	0.1	0.1	0.035	1.3	1.35	0.9	0.02	0.0215	20
20	0.1	0.15	0.05	0.9	1.6	1	0.025	0.0183	21
21	0.15	0.025	0.15	1.2	1.35	1	0.02	0.0211	22
22	0.15	0.035	0.025	1.3	1.6	1.1	0.025	0.02	20
23	0.15	0.05	0.035	0.9	1.85	1.2	0.03	0.0192	20
24	0.15	0.1	0.05	1	0.9	1.3	0.01	0.0197	22
25	0.15	0.15	0.1	1.1	1	0.9	0.015	0.0211	25

**Table 6 micromachines-14-02175-t006:** Optimization of process parameters.

Factor Classify	Factor A	Factor B	Factor C	Factor D	Factor E	Warpage Deformation (mm)	Cavitation
	Melt Temperature (Units: °C)	Mold Temperature (Units: °C)	Inject Time (Units: s)	Curing Time (Units: s)	Injection Pressure (Units: kg/cm^2^)		
Optimal Warpage	165	155	10	150	48	0.0170	22
Optimal Cavitation	155	175	12	170	42	0.00255	20
Optimal Comprehensive	155	185	10	170	36	0.0223	21
Initial	175	175	12	150	36	0.0203	26

**Table 7 micromachines-14-02175-t007:** Structure parameter optimization table (Units: mm).

Factor Classify	Factor A	Factor B	Factor C	Factor D	Factor E	Factor F	Factor G	Warpage Deformation	Cavitation
	Height of Gates	Height of Flow Channels	Height of Exhaust Ports	Width of Gates	Width of Flow Channels	Width of Exhaust Ports	Chamfer		
Optimal Warpage	0.15	0.15	0.035	1.2	1.6	1.3	0.025	0.0183	21
Optimal Cavitation	0.15	0.025	0.035	0.9	0.9	1.3	0.01	0.0423	19
Optimal Comprehensive	0.15	0.05	0.035	0.9	1.3	1.2	0.03	0.0192	20
Initial	0.04	0.04	0.05	1.17	1.1	1.1	—	0.0203	26

**Table 8 micromachines-14-02175-t008:** Table of material thermodynamic parameters.

Materials Characteristics	EK5600GHR	Lead Frame	Pb92.5Sn5Ag2.5	Si Chip
Density (g/cm^3^)	2	2	1.9	2.33
SHC (J/(kg·K))	1533	1095	1180	702
Thermal Conductivity (W/(m·K))	1.1	1.05	0.9	149
Thermal Expansivity (10^−6^/°C)	7 T < Tg 37 T > Tg	15.96	29	2.6
Young’s Modulus (MPa)	2927@175 °C 7580@130 °C 17,275@T < Tg	119,000	38,775	159,000
Poisson’s Ratio	0.3	0.33	0.3	0.25
Tg (°C)	120	—	—	—

**Table 9 micromachines-14-02175-t009:** Material wet parameter table.

Materials Characteristics	EK5600GHR	Lead Frame	Pb92.5Sn5Ag2.5	Si Chip
Csat(kg/m^3^)	5.455	1	1	1
Csat·D (kg/m·s)	3.273 × 10^−11^	1 × 10^−14^	1 × 10^−14^	1 × 10^−14^
Coefficient of Wet Expansion (mm^3^/mg)	0.00245475	—	—	—

**Table 10 micromachines-14-02175-t010:** Chip Size Table.

Size	Length	Width	Height
Classify			
Initial Size	1.75	1.3	0.2
1	1.95	1.4	0.2
2	2.15	1.5	0.2
3	2.35	1.6	0.2
4	2.55	1.7	0.2

**Table 11 micromachines-14-02175-t011:** The total deformation of moisture and heat superposition (Units: μm).

Classify	Initial Size	1	2	3	4
Location Number					
A	3.8566	3.9404	4.4539	3.9534	4.4721
B	3.7124	3.7614	3.7833	3.7794	3.7858
C	3.5702	3.6176	3.6687	3.6586	3.7132
D	4.4902	4.5177	6.1524	5.3179	6.2559
E	4.1303	4.2539	4.8508	4.3505	5.0002

## Data Availability

The data presented in this study are available on request from the corresponding author.

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
