# Peer review of "Study on Cavitation, Warpage Deformation, and Moisture Diffusion of Sop-8 Devices during Molding Process"

_micromachines, 2023, doi:10.3390/mi14122175_

Round 1

Reviewer 1 Report (Previous Reviewer 1)

Comments and Suggestions for Authors

#1. The quality of Figure 1 is very poor and requires improvement.

The rest is fine.

Author Response

Reviewer 2 Report (New Reviewer)

Comments and Suggestions for Authors

Plastic packaging of semiconductor chips received increasing interest due to its reduced carbon footprint during the manufacturing process. The article explores the challenges in plastic packaging of electronic devices, specifically focusing on reducing cavitation and warpage deformation in SOP-8 devices. This work aims to enhance the reliability of SOP-8 plastic packaging, a critical concern in the electronic industry, by proposing optimization strategies based on comprehensive simulations and experimental design. This work is interesting and significant but there exist several minor issues to be addressed before being considered for possible publication.

1. Data presentation requires improvement.

Figure 2 is difficult to see due to the poor resolution. The titles are misleading. Are both images captured from failed electronic devices? 

Figure 3 (b) is unclear.

There are non-English characters in Fig. 6(a), and the fonts are too small to read.

2. The authors need to provide clearer guidelines for industry application, possibly through case studies or collaborations with manufacturers.

3. Could benefit from more real-world testing to validate simulation results in future studies.

4. All acronyms should be defined at the first time they show up. For example, 'TQFP' on line 63 and "QFN" on line 65. Not recommend using 'Sop-8' unless it is a word that is really popular in the semiconductor community.

Comments on the Quality of English Language

1. English needs to be improved. There are language issues across the entire manuscript that should be fixed, given a few examples:

a. Line 52, 'Too many large cavitation will reduce the reliability of the device. As shown in Fig. 1 (a).' should be 'Too many large cavitations will reduce the reliability of the device, as shown in Fig. 1 (a).'

b. Line 63, TQFP' is an acronym that has never been defined, same as 'QFN' on Line 65.

c."...the most important is the injection moulding process"

etc.

Author Response

This manuscript is a resubmission of an earlier submission. The following is a list of the peer review reports and author responses from that submission.

Round 1

Reviewer 1 Report

Comments and Suggestions for Authors

Comment #1: I have read through the manuscript and therefore recommend the use of professional English Language editing tool(s) towards improving the quality of writing presented herewith.

Comment #2: What is the novelty of this article with regards to the adopted model?

Comment #3: The quality of figure 1 is extremely poor and must be replaced accordingly.

Comment #4: The tables relating to experimental designs may be moved to supplementary data.  The author(s) may state the process in the article itself and refer readers to the supplementary data. The results are the most important part that should be presented in the manuscript.

Comment #5: The clarity of the entire figures as well as the labels within the figures must be improved upon.

Comments on the Quality of English Language

I have read through the manuscript and therefore recommend the use of professional English Language editing tool(s) towards improving the quality of writing presented herewith.

Reviewer 2 Report

Comments and Suggestions for Authors

Overall review comments

The manuscript investigated process parameters and structural parameters in the process of plastic packaging. The moisture diffusion of plastic packaging devices is also studied. I consider the content of this manuscript will meet the reading interests of the readers of the journal. However, there are figure illustration issues, and the discussion and explanation should be further improved. Therefore, I suggest giving a major revision and the authors need to clarify some issues or supply more validation data to enrich the content.

1. The language requires major revision. The paper does not read smoothly (many typos) and it is redundant. I suggest the authors consult a native English speaker.

2. In the Introduction, the temperature effect of cavitation can be introduced, which

is a strategy for controlling the intensity of the cavitation. Kindly add the reference in the Introduction, of

-Polymer 217 (2021): 123428. https://doi.org/10.1016/j.polymer.2021.123428

- Int. J. Heat Mass Transfer 170 (2021): 120970, https://doi.org/10.1016/j.ijheatmasstransfer.2021.120970;

- "An Improved Mold Flow Optimization Technology for High-Density Power Modules."  23rd International Conference on Electronic Packaging Technology (ICEPT). IEEE, 2022. DOI: 10.1109/ICEPT56209.2022.9873411

- Ultrasonics Sonochemistry 86 (2022): 106035, https://doi.org/10.1016/j.ultsonch.2022.106035;

- Energy 254 (2022): 124426, https://doi.org/10.1016/j.energy.2022.124426;

- Journal of Cleaner Production (2022): 130470, https://doi.org/10.1016/j.jclepro.2022.130470.

3. The quality of Fig. 1 &2 needs to be improved. For apparatus, figures should be labelled and formatted correctly, with dimensions especially.  

4.The information shown in Fig. 3 is useless. Nothing can be found in this figure. Please revise it. The figures (almost all) are of poor quality.

5. Discussion and analysis for Results and Discussions in section 2&3 are not sufficient. What is the unique finding?

6. In section 3, Please give more details about the numerical settings.

7. Can you provide validation between the experimental and the numerical results?

8. Conclusion does not reflect the novelty of this work. Please make sure your conclusions’ section underscores the scientific value-added of your paper, and/or the applicability of your results. Highlight the novelty of your study. Clearly discuss what the previous studies that you are referring to are. What are the Research Gaps/Contributions?

9. The main issue of this manuscript is that it is more like a technical report of case studies. For a research paper, the mechanisms behind the phenomenon shall be carefully analyzed, identifying the main arguments, evidence, and conclusions. The goal is to develop a deep understanding of the research problem being investigated, the methods used to address it, and the key findings and implications for future research or practice.

Comments on the Quality of English Language

 The language requires major revision. The paper does not read smoothly (many typos) and it is redundant.

Reviewer 3 Report

Comments and Suggestions for Authors

This paper offers a thorough investigation into injection molding processes, structural parameters, and moisture diffusion in plastic packaging (Sop-8) of electronic devices. Numerical Simulations of the injection molding process are used to address cavitation and warpage deformation. The exploration of moisture diffusion in plastic packaging adds depth to the study.

However, I must express my reservations regarding the subject matter's novelty and the experimental results' adequacy. The manuscript lacks a clear articulation of the problem statement, which impacts the ensuing discussions. In various instances, the analysis comes across as superficial and unclear, failing to provide a cohesive narrative. In my assessment, the manuscript would greatly benefit from an enhanced analysis of the results, accompanied by a more comprehensive and insightful discussion. At present, the manuscript appears akin to a laboratory report detailing a specific case, rather than contributing to broader scientific discourse. One aspect that demands attention is the experimental characterization of the materials under investigation. In order to ensure the reproducibility and replicability of the tests, it is imperative that the manuscript incorporates essential details such as the geometrical dimensions of the molded part.

Moreover, the inclusion of complete constitutive models used to describe parameters such as viscosity, PVT (Pressure-Volume-Temperature), and moisture diffusion is essential to bolster the robustness of the study. Given the significance of the subject matter and the potential contributions this study can make to the field, I strongly recommend that the authors engage in substantial revisions. By addressing the highlighted concerns and aligning the manuscript with the standards of rigorous scientific inquiry, the authors can significantly enhance the quality and impact of their work.

Comments on the Quality of English Language

Moderate editing of English language required
